# Hysteresis Behavior Modeling of Magnetorheological Elastomers under Impact Loading Using a Multilayer Exponential-Based Preisach Model Enhanced with Particle Swarm Optimization

**DOI:** 10.3390/polym15092145

**Published:** 2023-04-30

**Authors:** Alawiyah Hasanah Mohd. Alawi, Khisbullah Hudha, Zulkiffli Abd. Kadir, Noor Hafizah Amer

**Affiliations:** Department of Mechanical Engineering, Faculty of Engineering, National Defence University of Malaysia, Kem Sungai Besi, Kuala Lumpur 57000, Malaysia; hasanahalawiyah@gmail.com (A.H.M.A.); zulkiffli@upnm.edu.my (Z.A.K.); noorhafizah@upnm.edu.my (N.H.A.)

**Keywords:** magnetorheological, elastomers, hysteresis behavior, impact loading, preisach model, particle swarm optimization, exponential function

## Abstract

Magnetorheological elastomers (MREs) are a type of smart material that can change their mechanical properties in response to external magnetic fields. These unique properties make them ideal for various applications, including vibration control, noise reduction, and shock absorption. This paper presents an approach for modeling the impact behavior of MREs. The proposed model uses a combination of exponential functions arranged in a multi-layer Preisach model to capture the nonlinear behavior of MREs under impact loads. The model is trained using particle swarm optimization (PSO) and validated using experimental data from drop impact tests conducted on MRE samples under various magnetic field strengths. The results demonstrate that the proposed model can accurately predict the impact behavior of MREs, making it a useful tool for designing MRE-based devices that require precise control of their impact response. The model’s response closely matches the experimental data with a maximum prediction error of 10% or less. Furthermore, the interpolated model’s response is in agreement with the experimental data with a maximum percentage error of less than 8.5%.

## 1. Introduction

MREs are smart materials that are composed of a polymer matrix filled with magnetic particles, usually iron or iron oxide, which can respond to an external magnetic field. The magnetic particles are typically dispersed within the elastomer matrix in a random manner, forming a network of interconnected chains [1]. When an external magnetic field is applied to the MRE, the magnetic particles within the polymer matrix align themselves with the direction of the field causing the elastomer to stiffen and become more rigid. Conversely, when the magnetic field is removed, the particles return to their random orientation and the elastomer returns to its original soft and flexible state [2,3]. The stiffness of MRE can be controlled by adjusting the strength and orientation of the magnetic field.

MREs have the potential for a range of applications in various industries. Some key examples include vibration control [4] and damping systems [5,6] where MREs can reduce vibrations and enhance the stability of structures and machinery. MREs can also be utilized in the development of soft robotics [7] and prosthetics with tunable stiffness [8], offering improved flexibility and control. Additionally, MREs have the potential for seismic protection [9], as they can enhance the seismic resistance of structures by providing adaptive damping to reduce the impact of seismic waves. In the aerospace and defense industries, MREs can be integrated into aircraft and spacecraft design to improve vibration control and damping [10], leading to better performance and stability. Furthermore, MREs may be applied to medical device design [11], such as stents and catheters, to offer greater control and flexibility. As research in this field progresses, the applications of MREs continue to expand, showcasing their potential for diverse applications in different industries [12].

MREs change their mechanical properties in response to an applied magnetic field. Due to their unique behavior, accurate modeling of MREs is crucial for their effective application in various industries. Accurate modeling of MREs allows for the prediction of their behavior under different magnetic field strengths which is essential in designing MRE-based devices and systems. It also enables the optimization of MREs for specific applications by predicting the mechanical properties, such as stiffness and damping, and the response times of MREs to changes in the magnetic field. Furthermore, accurate modeling of MREs can lead to the development of more advanced MRE-based systems that require a more precise understanding of the behavior of MREs to ensure optimal performance.

There are two methods for modeling the behavior of MREs, namely parametric and non-parametric models [13]. Parametric models refer to models that are based on a set of predefined parameters or assumptions about the behavior of the material. These models typically involve the use of mathematical equations and require knowledge of the material’s properties and characteristics, such as its stiffness and damping. On the other hand, non-parametric models do not rely on pre-defined assumptions or parameters but instead use data-driven approaches to develop models based on observed behavior. These models can include machine learning algorithms, such as artificial neural networks, or support vector machines that learn from data to predict the behavior of the material.

Both parametric and non-parametric models have been used in MRE research. Parametric models, such as micromechanical models or continuum mechanics models, have been developed to describe the behavior of MREs based on the physical properties of the material. Non-parametric models, such as artificial neural networks [14], or fuzzy logic systems [15], have been used to predict the behavior of MREs based on experimental data. Overall, both approaches have their strengths and weaknesses, and the choice of which model to use depends on the specific application and available data. Parametric models can provide a more fundamental understanding of the material behavior, but they may require more knowledge of the material properties. Non-parametric models can be more flexible and adaptable but may require more experimental data to develop accurate models [16,17].

In this paper, an impact behavior model for MREs that takes into account the hysteresis characteristics of MREs is proposed. To model the hysteresis characteristics, the multilayer exponential-based Preisach model is proposed. Exponential functions are commonly used in modeling hysteresis as they can capture the nonlinear behavior of the material. The multilayer exponential function allows for a more complex model that can better capture the behavior of MREs. To optimize the parameters of the model, an optimization tool, namely PSO, is used. PSO is a nature-inspired optimization algorithm that simulates the social behavior of swarms in nature, such as flocks of birds or schools of fish. The algorithm is designed to find the optimal solution to a given optimization problem by iteratively adjusting a group of particles or agents in a search space.

## 2. Design and Fabrication of MRE-Based Double Acting Actuator

In traditional MREs, the actuator works mainly in compression mode [18]. However, in this study, a double-acting MRE actuator (Appendix A) that can work in both compression and extension modes was developed and fabricated. The schematic diagram and the prototype of the MRE actuator are shown in Figure 1a,b, respectively. MRE actuators consist of a piston and cylinder containing MRE material, a coil or electromagnet, and a power source. When a mechanical vibration causes the piston to move relative to the cylinder, a magnetic field is generated by the coil or electromagnet which causes the magnetic particles in the MRE to align themselves in the direction of the field. This changes the stiffness and damping properties of the MRE, providing a damping force that opposes the motion of the piston. The damping force can be adjusted by varying the strength and direction of the magnetic field, which is controlled by the controller.

The fabrication process of MRE typically involves several steps. First, an elastomer matrix is selected to meet specific application requirements. Next, magnetic particles are selected and their size is determined. Then, the magnetic particles are prepared by mixing them with the elastomer. The mixture is cured by pouring it into a mold. Finally, the MRE is magnetized by exposing it to a strong magnetic field which magnetizes the particles and induces the desired mechanical properties [19,20]. The coils are made of copper wire coated with a layer of enamel insulation with a diameter of 0.7 mm; the number of coil turns is 250. The parameters and the composition for MRE fabrication are shown in Table 1.

## 3. Experimental Study on the Force-Displacement Characteristics of MREs for Impact Response

Upon fabricating the proposed MRE, a series of drop impact tests were conducted to evaluate the effectiveness of the MRE under impact loads by analyzing their force-displacement characteristics. The drop impact test is performed to assess the ability of a material to resist impact loading. The test involves several steps. First, the specifications for the test, such as the height of the drop, the weight of the impactor, and other relevant factors, are determined. Next, the test specimen, or product, is prepared to ensure that it is in a suitable condition for testing. Then the impactor is secured to the drop apparatus, carefully aligned, and oriented. The impactor is raised to the predetermined height and released, allowing it to free-fall and strike the test specimen. The impact is observed and to ensure accurate and reliable results, the test is typically repeated several times. Finally, the results can be analyzed. Figure 2 depicts the drop test machine that was utilized in the experiment.

The tests were carried out by subjecting the material to a sudden external force to observe its behavior. The Instron Drop Impact Machine and CEAST Software were utilized to set the experiment variables, including the impact energy, impact velocity, falling height, total mass, and applied current. The parameters set to the Instron Drop Impact Machine during the experiment are shown in Table 2. Figure 2 shows the experimental setup that was conducted in the Automotive Lab at Universiti Pertahanan Nasional Malaysia (UPNM).

The experiments were conducted to investigate the behavior of the MRE under different current inputs to the coils. In order to achieve this, varying current values were applied to the coils at 0, 0.5, 1, 1.5, and 2 Amperes. To ensure the accuracy and consistency of the data collected, each experiment was repeated multiple times at each current level. This approach helps to reduce the effects of random errors and improves the reliability of the experimental results. Figure 3 presents the experimental results displaying the force-displacement relationship with respect to the different current inputs. The figure clearly illustrates the changes in stiffness and damping properties of the MRE with varying current inputs. Upon analyzing each force-displacement curve, it was found that the upper slope of the curve represents the response of the MRE undergoing compression, while the lower slope represents the response of the MRE undergoing retraction.

## 4. Modeling the Hysteresis Characteristics of MREs Using a Multilayer Exponential-Based Preisach Model Optimized with PSO

This section describes the algorithm for a multilayer exponential-based Preisach model that was developed using experimental data on the force-displacement characteristics of MREs under impact loading conditions for input currents injected to the coils at 0, 0.5, 1, 1.5, and 2 Amperes. Additionally, the interpolation method used to obtain the force produced by the MREs for input currents between the specified values is explained. Finally, an optimization tool called Particle Swarm Optimization (PSO) is also discussed.

### 4.1. Multilayer Exponential-Based Preisach Model

The Preisach model is a mathematical tool used to describe hysteresis, which is a phenomenon in which the output of a system depends not only on the current input but also on its history [22,23]. The model consists of a set of hysterons, which are basic units that represent the behavior of the system. Each hysteron is associated with a particular input/output history and has a unique switching threshold. The model describes the behavior of the system by representing it as a distribution of hysterons, each of which contributes to the overall response of the system [24]. The Preisach model consists of many relay hysterons connected in parallel, given weights, and summed. This can be visualized by a block diagram as shown in Figure 4. Each of these relays has different *α* and *β* thresholds and is scaled by *µ*. By increasing the number of *N*, the true hysteresis curve can be better approximated [25].

An exponential function is a mathematical function in the form of $f\left(x\right)=e^{x}$. Exponential functions are used to model phenomena that grow or decay at a constant percentage rate over time, such as population growth, radioactive decay, or compound interest. They have a characteristic curve that starts rapidly and then increases or decreases slowly. First, second and *n*-th hysterons are defined as follows:$$\mu_{1}R_{\alpha_{1}\beta_{1}}=a_{1}e^{b_{1}x}$$
$$\mu_{2}R_{\alpha_{2}\beta_{2}}=a_{2}e^{b_{2}x}$$
…
$$\mu_{N}R_{\alpha_{N}\beta_{N}}=a_{N}e^{b_{N}x}$$

The general relationship between input-output of the Preisach model is written as follows:$$f\left(x\right)=a_{1}e^{b_{1}x}+a_{2}e^{b_{2}x}+\cdots +a_{N}e^{b_{N}x}\;$$

Referring to Figure 3, there are five hysteresis loops representing the force-displacement characteristics of MREs under impact loading for different amounts of current injected into the coils. The currents include 0, 0.5, 1, 1.5, and 2 Amperes. Each loop has two Preisach model setups representing the upper and lower slopes.

In this proposed model, some controlled parameters are optimized using PSO to accurately represent the hysteresis response of MREs under impact loadings. They are: $a_{1}$, $a_{2}$, …, $a_{N}$; $b_{1}$, $b_{2}$, …, $b_{N}$; $c_{1}$, $c_{2}$, …, $c_{N}$; $d_{1}$, $d_{2}$, …, $d_{N}$; …; $s_{1}$, $s_{2}$, …, $s_{N}$; $t_{1}$, $t_{2}$, …, $t_{N}$.
$$f{\left(x\right)}_{0}^{up}=a_{1}e^{b_{1}x}+a_{2}e^{b_{2}x}+\cdots +a_{N}e^{b_{N}x}$$
$$f{\left(x\right)}_{0}^{lo}=c_{1}e^{d_{1}x}+c_{2}e^{d_{2}x}+\cdots +c_{N}e^{d_{N}x}$$
$$f{\left(x\right)}_{0.5}^{up}=e_{1}e^{f_{1}x}+e_{2}e^{f_{2}x}+\cdots +e_{N}e^{f_{N}x}$$
$$f{\left(x\right)}_{0.5}^{lo}=g_{1}e^{h_{1}x}+g_{2}e^{h_{2}x}+\cdots +g_{N}e^{h_{N}x}$$
$$f{\left(x\right)}_{1}^{up}=i_{1}e^{j_{1}x}+i_{2}e^{j_{2}x}+\cdots +i_{N}e^{j_{N}x}$$
$$f{\left(x\right)}_{1}^{lo}=k_{1}e^{l_{1}x}+k_{2}e^{l_{2}x}+\cdots +k_{N}e^{l_{N}x}$$
$$f{\left(x\right)}_{1.5}^{up}=m_{1}e^{n_{1}x}+m_{2}e^{n_{2}x}+\cdots +m_{N}e^{m_{N}x}$$
$$f{\left(x\right)}_{1.5}^{lo}=o_{1}e^{p_{1}x}+o_{2}e^{p_{2}x}+\cdots +o_{N}e^{p_{N}x}$$
$$f{\left(x\right)}_{2}^{up}=q_{1}e^{r_{1}x}+q_{2}e^{r_{2}x}+\cdots +q_{N}e^{r_{N}x}$$
$$f{\left(x\right)}_{2}^{lo}=s_{1}e^{t_{1}x}+s_{2}e^{t_{2}x}+\cdots +s_{N}e^{t_{N}x}$$

As previously mentioned, the proposed model was developed based on experimental data obtained from the force-displacement measurements for input currents of 0, 0.5, 1, 1.5, and 2 Amperes. To calculate the force generated by the MREs for input currents between 0 A and 0.5 A, an interpolation approach was used based on the known or specified displacement. The interpolation algorithm is detailed in Figure 5, which uses the experimental data for 0 Ampere and 0.5 Ampere as an example. Similar algorithms are used to calculate the force generated by MREs for input currents between 0.5 A and 1 A, between 1 A and 1.5 A, and between 1.5 A and 2 A. The algorithm for calculating force produced by MREs is as follows:$$F{\left(x\right)}_{i}^{up}=F{\left(x\right)}_{0}^{up}+\left(F{\left(x\right)}_{0.5}^{up}-F{\left(x\right)}_{0}^{up}\right)i$$
$$F{\left(x\right)}_{i}^{lo}=F{\left(x\right)}_{0}^{lo}+\left(F{\left(x\right)}_{0.5}^{lo}-F{\left(x\right)}_{0}^{lo}\right)i$$
where

i: Current injected to the MREs coils (between 0 A to 0.5 A)

$F{\left(x\right)}_{i}^{up}$: Force produced by MRE at current *i* during compression

$F{\left(x\right)}_{i}^{lo}$: Force produced by MRE at current *i* during retraction

$F{\left(x\right)}_{0}^{up}$: Force at upper slope of 0 A

$F{\left(x\right)}_{0}^{lo}$: Force at lower slope of 0 A

$F{\left(x\right)}_{0.5}^{up}$: Force at upper slope of 0.5 A

$F{\left(x\right)}_{0}^{lo}$: Force at lower slope of 0.5 A

### 4.2. Optimization of Multilayers Sigmoidal Functions Using PSO

PSO is a metaheuristic optimization algorithm that is inspired by the social behavior of bird flocking or fish schooling [26]. In PSO, a set of particles are initialized randomly in the search space and move towards the optimal solution by updating their positions based on their own previous positions and the positions of the best-performing particles in the swarm. The basic theory of PSO can be summarized in the following steps [27]:Initialization: A population of particles is generated randomly in the search space. Each particle represents a potential solution to the optimization problem.Evaluation: The fitness of each particle is evaluated based on the objective function to be optimized.Update of the particle’s best position: Each particle keeps track of the best position it has visited so far, denoted as *P_best_*. If the fitness of the current position is better than its *P_best_*, the particle updates its *P_best_*.Update of the swarm’s best position: The best position among all the *P_best_* positions of the particles is denoted as *G_best_*. If the fitness of the current *G_best_* position is better than its previous value, the swarm updates its *G_best_*.Update of the particle’s velocity and position: Each particle updates its velocity and position based on its current velocity, its distance from its *P_best_*, and its distance from the *G_best_*. The velocity and position updates are given by the following equations:

Velocity update:$$V_{\left(t+1\right)}^{i}=w\cdot V_{\left(t\right)}^{i}+c_{1}\cdot r_{1}\cdot \left(P_{best}^{i}-X_{\left(t\right)}^{i}\right)+c_{2}\cdot r_{2}\cdot \left(G_{best}-X_{\left(t\right)}^{i}\right)$$

Position update:$$X_{\left(t+1\right)}^{i}=X_{\left(t\right)}^{i}+V_{\left(t+1\right)}^{i}$$
where $V_{\left(t\right)}^{i}$ and $X_{\left(t\right)}^{i}$ are the velocity and position of particle *i* at time *t*, *w* is the inertia weight, $c_{1}$ and $c_{2}$ are the acceleration coefficients, and $r_{1}$ and $r_{2}$ are random numbers between 0 and 1.

6.Termination: The algorithm terminates when a stopping criterion is met, such as reaching a maximum number of iterations or a satisfactory fitness level.7.By iterating through these steps, the particles in the swarm collectively move toward the optimal solution of the optimization problem.

In this study, the parameters of the multilayer exponential-based Preisach model that will be tuned using PSO are $a_{1}$, $a_{2}$, …, $a_{N}$; $b_{1}$, $b_{2}$, …, $b_{N}$; $c_{1}$, $c_{2}$, …, $c_{N}$; $d_{1}$, $d_{2}$, …, $d_{N}$; …; $s_{1}$, $s_{2}$, …, $s_{N}$; $t_{1}$, $t_{2}$, …, $t_{N}$. In PSO, there are several optimization parameters that need to be set to ensure the algorithm performs optimally. The choice of parameter values for PSO can depend on the specific problem being solved and the characteristics of the search space. A common approach is to use a parameter tuning method, such as grid search or random search, to find the optimal parameter values. These parameters include:Swarm size: 40Maximum number of iterations: 100Inertia weight: 0.9Acceleration coefficients ($c_{1}$ and $c_{2}$): 1.42

## 5. Results and Discussions

This section presents a comparison between the simulated responses of the proposed MRE model and the corresponding experimental data. The maximum error of the predicted force will also be analyzed and discussed. Finally, the accuracy of the model will be tested and validated using input currents ranging from 0 to 0.5 A, 0.5 to 1 A, 1 to 1.5 A, and 1.5 to 2 A. Effects of varying the swarm size and the number of iterations on the PSO are also analyzed.

### 5.1. Comparison between the Simulated Response of Multilayer Exponential-Based Preisach Model with the Experimental Data

Figure 6 compares the simulated model response to experimental data where no current was injected into the coils. The model’s response closely matches the experimental data, indicating a high degree of accuracy in the simulation. However, a maximum error of 5% occurred in area A, which may be attributed to limitations in the experimental setup or inaccuracies in the simulation model. Further investigation and refinement of the model could potentially reduce this error and improve the overall accuracy of the simulation.

In Figure 7, the simulated model response is compared to experimental data for an input current of 0.5 A injected into the coils. The simulation results exhibit a high degree of accuracy and closely match the experimental data, but the maximum error occurs in area B with a percentage error of approximately 10%. This discrepancy could be due to limitations in the experimental setup, variations in the material properties, or uncertainties in the simulation model. Despite this limitation, the close agreement between the simulated and experimental data validates the model’s capability to predict the system’s behavior accurately under different input conditions.

Figure 8 compares the simulated model response to experimental data for an input current of 1 A injected into the coils. The simulation results exhibit a high degree of accuracy and closely match the experimental data, but the maximum error is observed in area C with a percentage error of approximately 4.8%. Again, this discrepancy may be due to experimental limitations, material property variations, or uncertainties in the simulation model. However, the close agreement between the simulated and experimental data provides confidence in the model’s ability to accurately predict the system’s behavior.

Figure 9 compares the simulated model response to experimental data for the input current of 1.5 A injected into the coils. The simulation results closely match the experimental data, demonstrating the model’s high degree of accuracy in predicting the system’s response under this input condition. However, it is worth noting that the maximum error occurs in area D, with a percentage error of approximately 4.7%. The close agreement between the simulated and experimental data validates the model’s capability to accurately predict the system’s behavior.

Lastly, in Figure 10, the simulated model response is compared to experimental data for an input current of 2 A injected into the coils. The simulation results closely match the experimental data, indicating that the model accurately predicts the system’s response under this input condition. However, the maximum error is observed in area E with a percentage error of approximately 8.1%. Nevertheless, the close agreement between the simulated and experimental data validates the model’s capability to predict the system’s behavior accurately. The maximum error of the predicted force of the proposed model is summarized in Table 3.

### 5.2. Validation of the Interpolated Model

Previously, it was mentioned that the force-displacement characteristics of MREs under impact loading for input currents ranging from 0–0.5, 0.5–1, 1–1.5, and 1.5–2 Amperes were predicted using an interpolation algorithm. This algorithm is commonly used to estimate data points within a range of values based on known data points. To validate the accuracy of this interpolation algorithm, experimental works were conducted using drop impact tests. These tests were carried out with the input currents injected into MRE coils of 0.3, 0.7, 1.3, and 1.7 Amperes, and the resulting force-displacement data were recorded. The experimental data obtained from the drop impact tests were then compared to the corresponding model response with the same input current as shown in Figure 11. It can be seen from the figure that the interpolation algorithm is valid. The model responses closely follow the experimental results with an acceptable error. The maximum error of the predicted force of the proposed model in the interpolation regions is summarized in Table 4.

### 5.3. Effects of Varying Swarm Size and the Number of Iterations

The number of iterations is an important parameter in PSO, as it determines the length of time the particles are allowed to search for the optimal solution. In general, increasing the number of iterations in PSO can improve the algorithm’s ability to find the global optimum, as it allows the particles more time to explore the search space and converge on the best solution. However, at a certain point, additional iterations may not lead to any further improvement in the solution, as the particles may have already converged to a local optimum. On the other hand, decreasing the number of iterations can lead to faster execution times, but at the cost of potentially missing out on better solutions. Therefore, it is important to strike a balance between the number of iterations and the desired level of performance index and execution time. Referring to Figure 12, the number of iterations selected was 100 since it has a fast convergence rate and is able to achieve a performance index as good as the performance index for 120 iterations.

The swarm size refers to the number of particles in the population and it determines the diversity and convergence rate of the swarm. Increasing the swarm size can lead to better global exploration, as there are more particles searching the solution space. However, it can also increase the computational cost and reduce the convergence rate, as there are more particles to communicate and update. On the other hand, reducing the swarm size can improve the convergence rate, as there are fewer particles to communicate and update. However, it may also decrease the diversity of the swarm, which can lead to premature convergence and suboptimal solutions. Referring to Figure 13, the swarm size selected in this study was 40 as it shows a fast convergence rate and achieved a performance index as good as a swarm size of 100.

## 6. Conclusions

This paper presents a comprehensive study on the hysteresis behavior modeling of magnetorheological elastomers under impact loadings. The study proposes a multilayered exponential-based Preisach model that is enhanced with particle swarm optimization to provide a reliable and accurate framework for capturing the complex hysteresis behavior of the material. The developed model demonstrates excellent performance in capturing the dynamic response of magnetorheological elastomers under various impact-loading scenarios. The results show that the model’s response closely matches the experimental data, with a maximum prediction error of 10% or less. The interpolated model’s response also shows good agreement with the experimental data, with a maximum percentage error of less than 8.5%. The study also examines the effects of varying the number of iterations and the number of particles on the performance of PSO. Overall, the findings suggest that the proposed model provides a promising approach for accurately predicting the hysteresis behavior of magnetorheological elastomers under impact loadings.

## Figures and Tables

**Figure 1 polymers-15-02145-f001:**
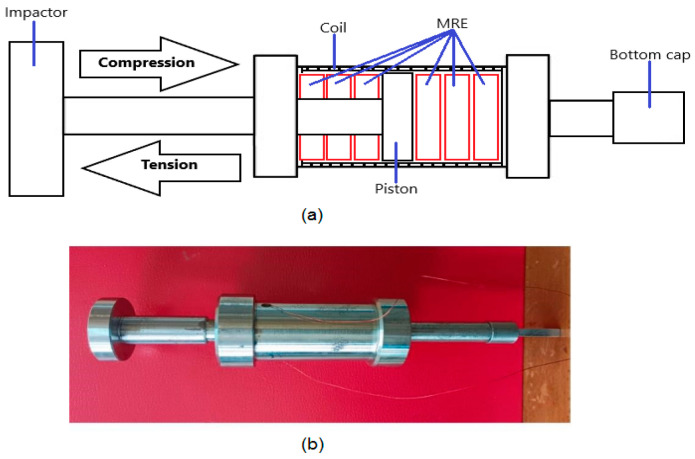
Schematic Diagram of a Double Acting MRE. The schematic diagram (**a**) and the prototype (**b**) of the MRE actuator.

**Figure 2 polymers-15-02145-f002:**
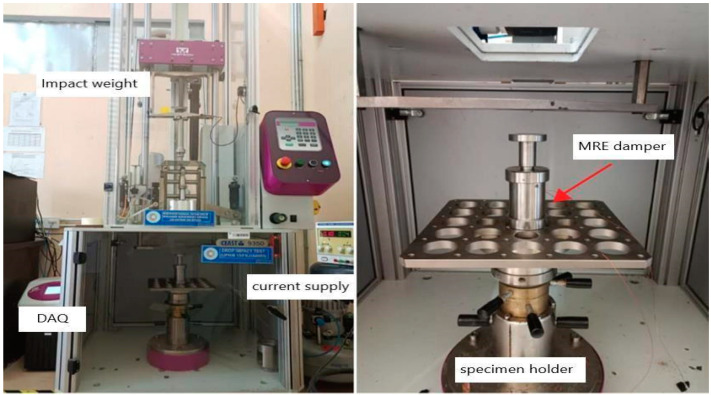
Drop Impact Test Machine used in this Study.

**Figure 3 polymers-15-02145-f003:**
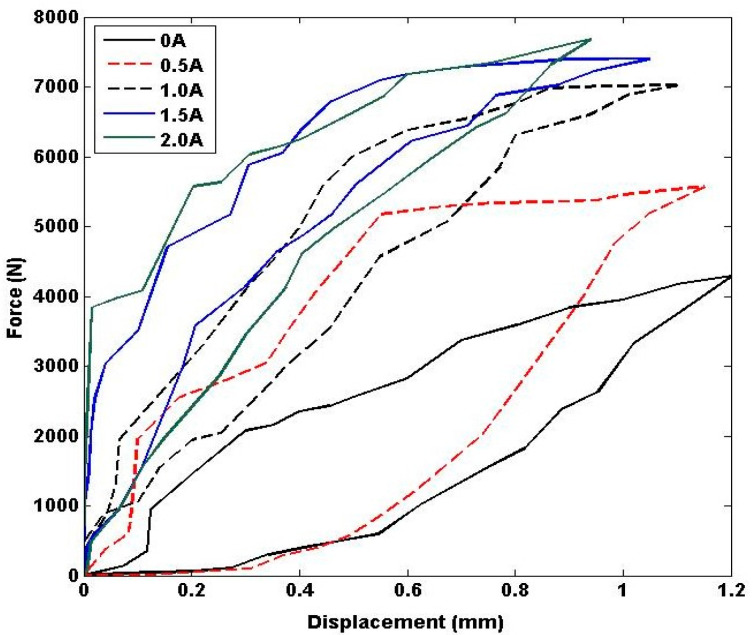
Force-Displacement Characteristics of MREs in Various Currents.

**Figure 4 polymers-15-02145-f004:**
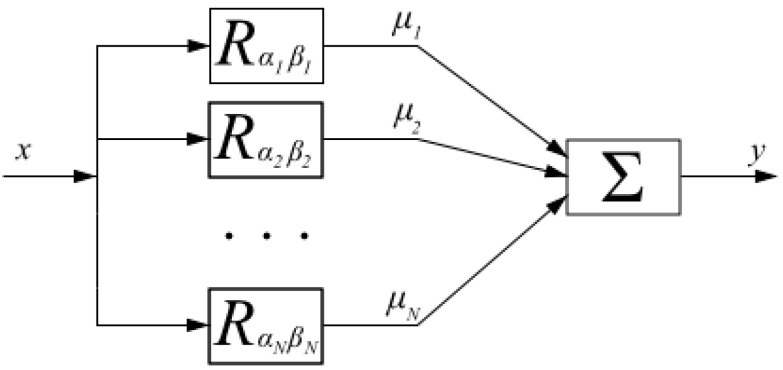
Block Diagram of Preisach Model.

**Figure 5 polymers-15-02145-f005:**
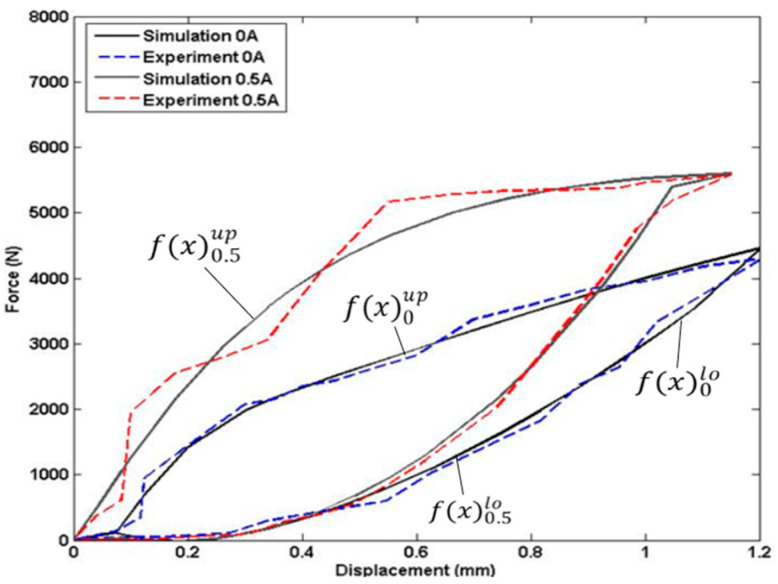
Interpolation of Force-Displacement Characteristics for Input Current Between 0–0.5 Ampere.

**Figure 6 polymers-15-02145-f006:**
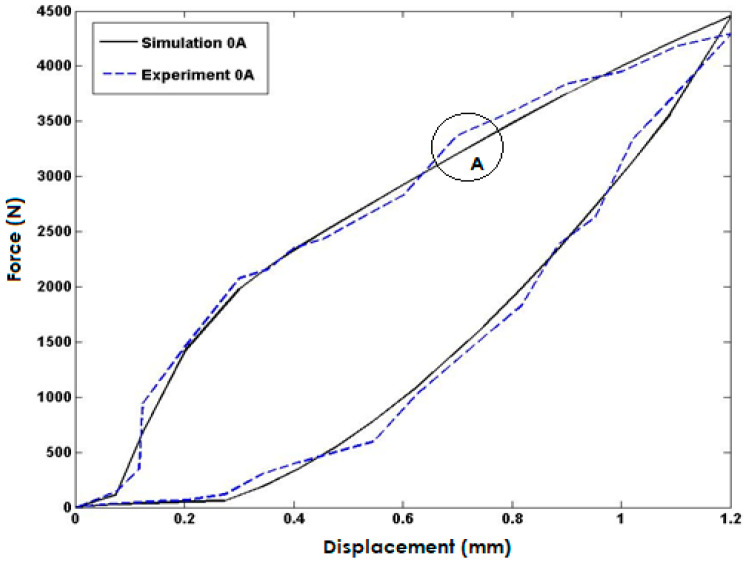
Comparisons with the Model Response and Experimental Data for the Input Current of 0 Ampere.

**Figure 7 polymers-15-02145-f007:**
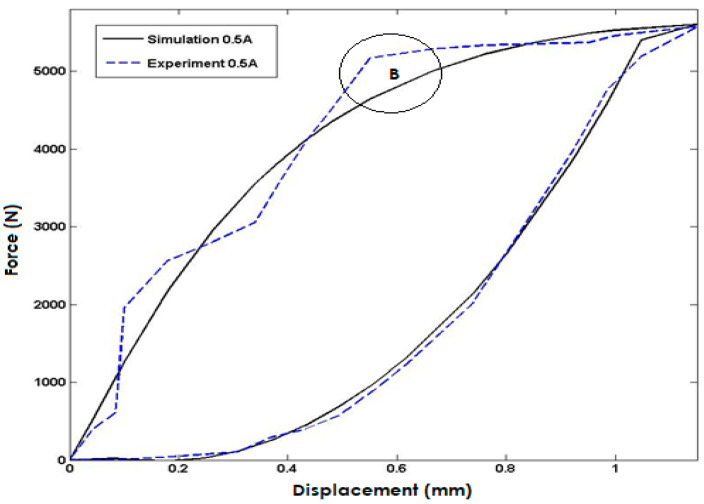
Comparisons with the Model Response and Experimental Data for the Input Current of 0.5 Ampere.

**Figure 8 polymers-15-02145-f008:**
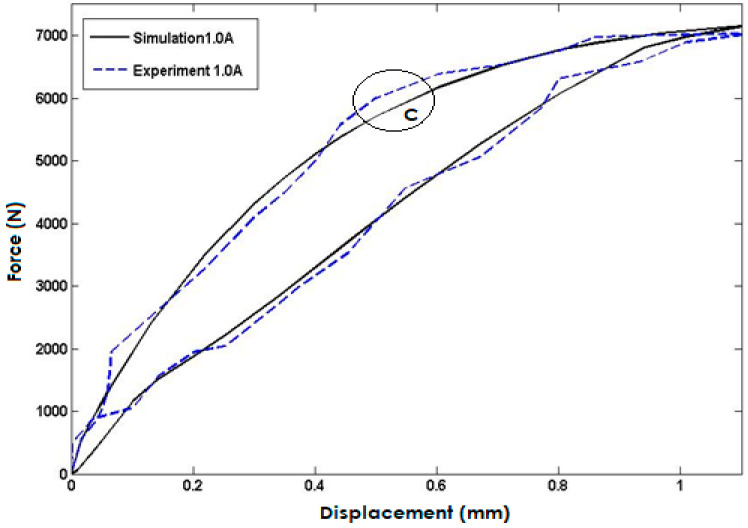
Comparisons with the Model Response and Experimental Data for the Input Current of 1 Ampere.

**Figure 9 polymers-15-02145-f009:**
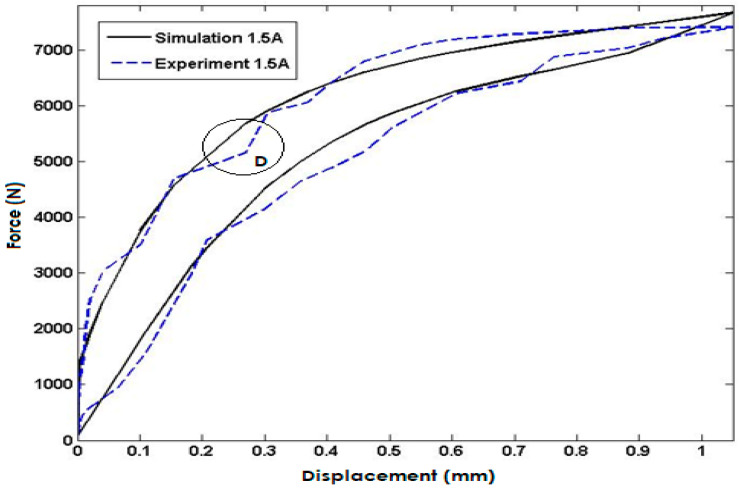
Comparisons with the Model Response and Experimental Data for the Input Current of 1.5 Ampere.

**Figure 10 polymers-15-02145-f010:**
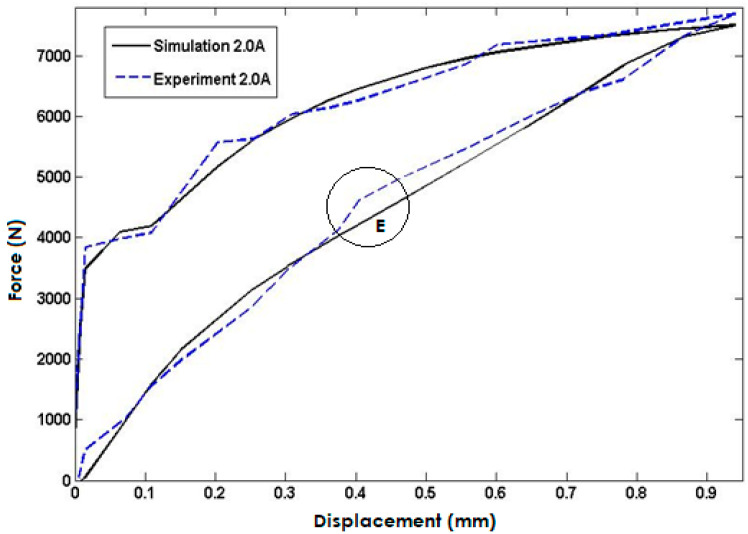
Comparisons with the Model Response and Experimental Data for the Input Current of 2 Ampere.

**Figure 11 polymers-15-02145-f011:**
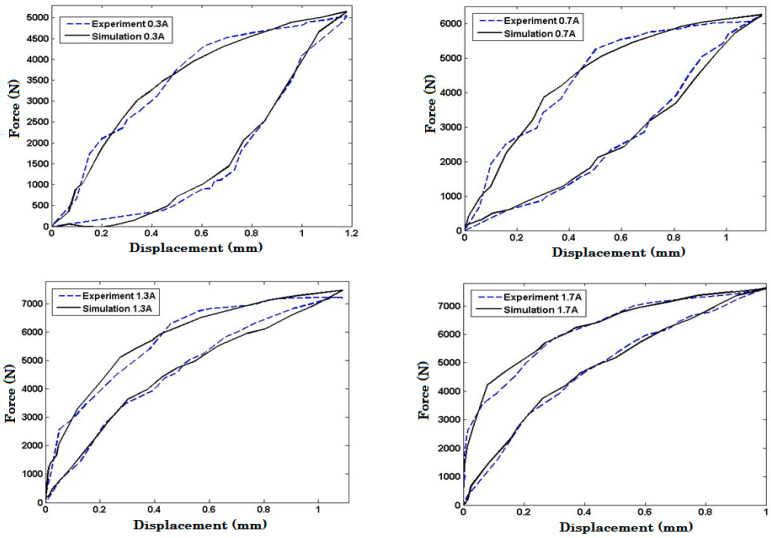
Validation of the Interpolated Model in Various Input Currents.

**Figure 12 polymers-15-02145-f012:**
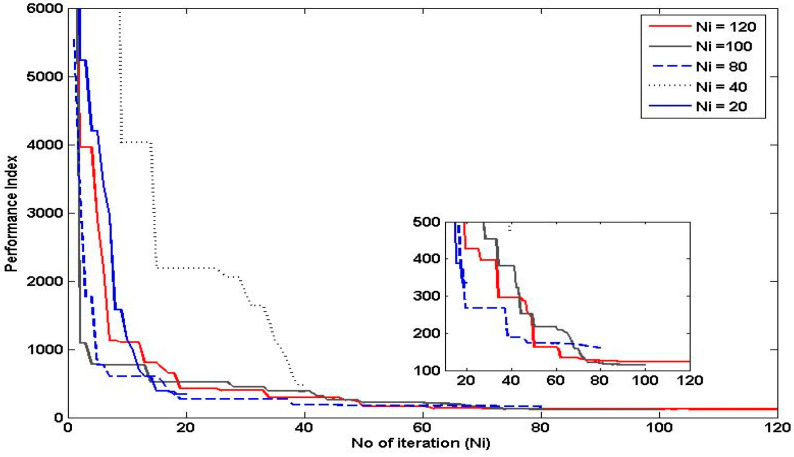
Effects of Varying the Number of Iterations on the Performance of PSO.

**Figure 13 polymers-15-02145-f013:**
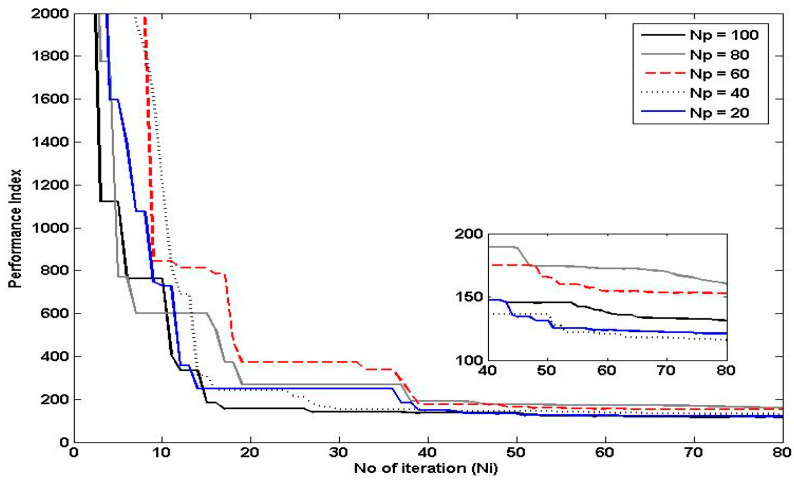
Effects of Varying the Number of Particles on the Performance of PSO.

**Table 1 polymers-15-02145-t001:** Composition of the MRE Sample [21].

Materials	Percentage
RTV Silicone Rubber	30%
Carbonyl Iron Powder	60%
Additive (Ferrite)	7%
Hardener (Isocyanates)	3%

**Table 2 polymers-15-02145-t002:** Parameters of The Drop Impact Test.

Parameters	Input Values
Impact Energy	13.8 J
Impact Velocity	2.24 m/s
Falling Height	256 mm
Total Mass	5.5 kg
Current	0–2 A
Contact area	13.87 cm^2^

**Table 3 polymers-15-02145-t003:** The Maximum Percentage of Error in All Five Cases.

Current	Experimental Data	Simulation Data	Percentage of Error (%)
0 A	3374.804	3206.756	4.98
0.5 A	5171.588	4642.620	10.23
1.0 A	6008.422	5718.536	4.83
1.5 A	7093.411	6762.593	4.66
2.0 A	5572.439	5123.559	8.1

**Table 4 polymers-15-02145-t004:** The Maximum Percentage of Error for the Interpolated Model.

Current	Experimental Data	Simulation Data	Percentage Error (%)
0.3 A	4321.450	3954.718	8.486
0.7 A	5263.420	5072.986	3.618
1.3 A	6746.820	6518.139	3.389
1.7 A	4982.910	4707.04	5.536

## Data Availability

Not applicable.

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
