# Peer review of "Hysteresis Behavior Modeling of Magnetorheological Elastomers under Impact Loading Using a Multilayer Exponential-Based Preisach Model Enhanced with Particle Swarm Optimization"

_polymers, 2023, doi:10.3390/polym15092145_

Round 1
Reviewer 1 Report
This paper presents an approach for impact behavior if magnetorheological elastomers (MREs) with arranged a multi-layer Preisach model which is trained using particle swarm optimization. MREs are attractive material for various applications in response to external magnetic fields. The proposed model predicted the motion of MREs with impact loads with reasonable accuracy, however, it is better to describe especially for the advantages of using this model and the experimental details for data collection. I cannot recommend publishing an article in its current form in Polymers until the author clarifies at least the points listed below.
1) On page 3, in Figure 1, it is worth to describe a size of the fabricated double action MRE for considering size effect and reproducibility of measurements. It is also better to indicate the source of materials used.
2) On page 4, in Table 2, it is better to specify the contact area of the Instron drop impact machine to evaluate impact per area.
3) On page 7, in Figure 5, it might be both solid lines with distinct colors in the legend for 0 A and 0.5 A force vs displacement curves. Solid and dotted line express calculated and experimental values.
Author Response
Please see the attacment...

Reviewer 2 Report
Hysteresis Behavior Modeling of Magnetorheological Elastomers under Impact Loadings using MultiLayers Exponential based Preisach Model Enhanced with Particle Swarm Optimization.
Very interesting topic to apply research here.
I would like to star my review of your work by asking the following: What is the value added of YOUR work to what has been previously investigated regarding magnetorheological elastomers?
There is a good approach in your introduction section, What type of applications are readily available in the market? one very importan could be the suspension components for automotive industry...
For the testings you performed, how many samples or replicas did you run? what is the standard deviation of your experiments?
in your work, you mentioned "The developed model demonstrates excellent performance in capturing the dynamic response of magnetorheological elastomers under various impact loading scenarios.", the correlation of experiments and the model is important here, dynamic response of the MRE under diverse mechanical scenarios for sure is interesting to the readers, please detail more on this.
I would recommend to search and use more recent references (>2021 at least)
English grammar seems good to me.
